# A Pyrrhic Victory: The PMN Response to Ocular Bacterial Infections

**DOI:** 10.3390/microorganisms7110537

**Published:** 2019-11-07

**Authors:** Erin T. Livingston, Md Huzzatul Mursalin, Michelle C. Callegan

**Affiliations:** 1Department of Microbiology and Immunology, The University of Oklahoma Health Sciences Center, Oklahoma City, OK 73104, USA; erin-livingston@ouhsc.edu (E.T.L.); MDHuzzatul-Mursalin@ouhsc.edu (M.H.M.); 2Department of Ophthalmology, The University of Oklahoma Health Sciences Center, Oklahoma City, OK 73104, USA; 3Oklahoma Center for Neuroscience, The University of Oklahoma Health Sciences Center, Oklahoma City, OK 73104, USA; 4Dean McGee Eye Institute, Oklahoma City, OK 73104, USA

**Keywords:** polymorphonuclear leukocytes, neutrophils, innate immunity, bacteria, keratitis, conjunctivitis, endophthalmitis, uveitis

## Abstract

Some tissues of the eye are susceptible to damage due to their exposure to the outside environment and inability to regenerate. Immune privilege, although beneficial to the eye in terms of homeostasis and protection, can be harmful when breached or when an aberrant response occurs in the face of challenge. In this review, we highlight the role of the PMN (polymorphonuclear leukocyte) in different bacterial ocular infections that invade the immune privileged eye at the anterior and posterior segments: keratitis, conjunctivitis, uveitis, and endophthalmitis. Interestingly, the PMN response from the host seems to be necessary for pathogen clearance in ocular disease, but the inflammatory response can also be detrimental to vision retention. This “Pyrrhic Victory” scenario is explored in each type of ocular infection, with details on PMN recruitment and response at the site of ocular infection. In addition, we emphasize the differences in PMN responses between each ocular disease and its most common corresponding bacterial pathogen. The in vitro and animal models used to identify PMN responses, such as recruitment, phagocytosis, degranulation, and NETosis, are also outlined in each ocular infection. This detailed study of the ocular acute immune response to infection could provide novel therapeutic strategies for blinding diseases, provide more general information on ocular PMN responses, and reveal areas of bacterial ocular infection research that lack PMN response studies.

## 1. Introduction

In the 1940s, the unresponsive nature of the ocular immune environment was recognized by Sir Peter Medawar, who observed that foreign tissue grafts were not rejected when placed in the anterior chamber (AC) of the eye [1]. Medawar called this special relationship between the eye and the immune system “immune privilege”. Nearly eight decades of research on immune privilege has highlighted its highly intricate character, which is the result of highly coordinated interactions between multiple factors and mechanisms. One of these mechanisms is the efficient blood–retina barrier, which prevents the unrestricted entry and exit of cells and large molecules into and out of the eye. The eye also has a deficiency of efferent lymphatics, which also contributes to restricting infiltrating immune cells. Another contributing factor to immune privilege is the inhibitory ocular microenvironment of the eye. This unique environment consists of cell-bound and soluble immunosuppressive factors, which inhibits the recruitment and activity of immune cells. Involved in these processes are surface-bound molecules, including CD86, FasL, thrombospondin, and galectins [2,3]. Soluble factors include transforming growth factor-beta (TGF-β), calcitonin gene-related peptide (CGRP, a neuropeptide), and vasoactive intestinal peptide (VIP) [4,5]. Indeed, several mechanisms of immunosuppression and immunoregulation are utilized by the eye to establish and preserve immune privilege.

Other factors such as complement system proteins, antimicrobial peptides, and resident immune cells contribute to destroying pathogens without damaging ocular tissue. For example, tear fluid contains various antibacterial substances (lysozyme, lactoferrin, and surfactant protein D) secreted by lacrimal gland cells and ocular surface epithelial cells [6,7]. Complement in the vitreous is important in host defense against bacterial corneal infections [8], but may not be important in intraocular infections [9]. Soluble factors can be secreted by ocular resident cells, and these cells can also directly inhibit immune cells by contact-dependent mechanisms. For instance, the pigmented epithelia of the retina (RPE) and the iris or ciliary body have been shown to inhibit immune cell infiltration and induce T cells to become T regulatory cells [2]. However, in the face of severe infection, ocular immunosuppression cannot always effectively keep all immune cells from infiltrating and responding. This may be due to dysfunction of the protective blood–retina barrier caused by the pathogen, resulting in an infiltration of non-resident immune cells. These non-resident cells can cause irreversible tissue damage. Importantly, polymorphonuclear leukocytes (PMNs) have been consistently reported as one of the first responders and the most predominant cell type infiltrating into the eye when infection occurs [10,11].

More than 10^11^ PMNs, or neutrophils, are produced every day in the bone marrow, and these cells represent approximately 70% of all leukocytes [12]. With this impressive amount of production, it is no surprise that PMNs are also the most abundant white blood cell type in the human blood. Once in the blood, PMNs are trafficked to sites of infection. This process is known as the leukocyte adhesion cascade [13]. During this process, endothelial cells are activated by inflammatory chemokines or other chemoattractant factors released by cells affected during inflammation. When activated, endothelial cells express adhesion molecules on the luminal surface, such as intracellular adhesion molecule 1 (ICAM1) and vascular cell adhesion molecule 1 (VCAM1), causing PMN arrest and transmigration through the endothelial layer. PMNs display a different phenotype from the time they enter circulation to the time they migrate across the endothelial layer to a site of infection. This shift in phenotype is caused by transcriptional activation that is mediated, in part, by local granulocyte colony-stimulating factor (G-CSF) production and by changes in surface molecule expression or activity regulated by inflammatory factors at the infection site [14].

When PMNs recognize a pathogen, these cells utilize different functions and responses to clear the infection [14,15,16]. Phagocytosis involves the consumption of the organism into a phagocytic vacuole. This vacuole becomes a phagolysosome in which the organism is destroyed by low pH and destructive enzymes. PMNs also degranulate, releasing granules that contain a plethora of antimicrobial enzymes, into the extracellular environment. However, when the organism is too large to be consumed, PMNs can ensnare the pathogen by producing neutrophil extracellular traps (NETs). NETs are a tangle of DNA fibers and proteins released from PMN granules. In addition, PMNs can acquire specialized functions depending on their microenvironment [16,17]. PMNs, which are recruited into different areas of the infected eye (Figure 1), use a combination of these functions to fight a variety of bacterial ocular infections. Very similar to a “Pyrrhic Victory” scenario, PMN responses are necessary for clearing pathogens in the eye, but, as this review will highlight, the process of these responses has the unfortunate side effect of damaging and scarring delicate ocular tissues and threatening sight.

## 2. Conjunctivitis

Conjunctivitis results from inflammation of the conjunctiva, which is the transparent, lubricating mucous membrane that covers the anterior of the eye. The cause of this inflammation can be due to infectious or non-infectious agents. Conjunctivitis can also result from an aberrant proliferation of the conjunctival flora [18]. The result of an infection of this tissue is injection or dilation of the conjunctival vessels. Thus, the classic discharge, redness or hyperemia, and edema of the conjunctiva occurs (Figure 2). This infection can affect people of any age and demographic, but the disease is easily treatable and usually self-limiting. While these infections can be painful, they are typically responsive to topical antibiotics [19]. Most bacterial conjunctivitis patients receiving proper treatment recover with little to no change in visual acuity [20,21,22].

Bacterial conjunctivitis can spread by direct contact and has high transmission rates. The incidence of bacterial conjunctivitis is estimated to be 135 in 10,000 [21]. Transmission routes include the spread of fomites via contaminated fingers or oculogenital spread [22,23]. Predisposing factors for bacterial conjunctivitis include aberrant tear production, epithelial barrier disruption or dysfunction, trauma, and immunosuppression [24]. A large infiltration of inflammatory cells occurs during this infection. An infiltration of PMNs suggests a bacterial infection, while an infiltration of lymphocytes indicates viral and allergic conjunctivitis [25]. In adults, the most common bacterial pathogens for conjunctivitis are *Chlamydia trachomatis*, staphylococci, and *Streptococcus pneumoniae* [18,22,23,24,26].

### 2.1. Staphylococcus aureus Conjunctivitis

*Staphylococcus aureus* causes many infections including brain abscesses, osteomyelitis, pneumonia, septicemia, and skin infections [22,29,30]. This important human pathogen is a Gram-positive coccus, and is a leading cause of many ocular infections as well, including conjunctivitis, endophthalmitis, and keratitis [29,30,31,32,33]. To treat *S. aureus* conjunctivitis, empiric broad-spectrum antibiotic therapy typically shortens the recovery time and lessens the symptoms. However, Hautala et al. [34] reported that this infection is becoming more difficult to treat with the discovery that *S. aureus* isolates are increasingly methicillin-resistant (MRSA).

The infiltration and presence of PMNs during *S. aureus* conjunctivitis has been well described [25,35,36,37]. McCormick et al. [35] showed that the amount of infiltration and localization of PMN in the rabbit conjunctiva was dependent on the virulence of the infecting *S. aureus* strain being used. This model is particularly difficult to replicate due to the ability of the conjunctiva to rapidly recover with no apparent tissue damage. In this model, well-characterized laboratory strains grew slowly, resulting in a more localized PMN presence in the conjunctiva. In contrast, an ocular isolate obtained from a rabbit replicated much faster, resulting in a more significant PMN infiltration into the conjunctiva [35]. Zaidi et al. [37] reported that PMN infiltration may be dependent on the expression of surface/capsular polysaccharide poly-N-acetyl glucosamine (PNAG) on *S. aureus* and *Streptococcus pneumoniae*. Blocking PNAG with an opsonic antibody reduced the number of PMN infiltrating into the conjunctiva [37]. These studies demonstrated how certain staphylococcal factors affected the infiltration and localization of PMN in conjunctivitis. However, it is clear that studies on the role of the PMN and their function in *S. aureus* conjunctivitis are lacking. Studies on PMN function during *S. aureus* conjunctivitis will provide a clearer understanding of pathology for this disease.

### 2.2. Streptococcus pneumoniae Conjunctivitis

*S. pneumoniae* is a common cause of ocular surface infections [22,38,39]. This disease is associated with occurrences involving people in close living quarters, including university and military dormitories, daycare facilities, and special care facilities [38,39,40,41]. *S. pneumoniae* is one of the most common causative agents of acute conjunctivitis in children [42]. A majority of these outbreaks are caused by non-encapsulated strains [38,43]. The capsule of *S. pneumoniae* allows the bacteria to evade phagocytosis and killing, and the capsule is a known virulence factor in pneumonia and bacteremia [44,45,46]. Inflammation and the presence of PMNs during experimental *S. pneumoniae* conjunctivitis have been observed [47]. However, just as with *S. aureus* conjunctivitis studies, the study of the specific roles of PMNs during this infection is lacking.

Perhaps the most in-depth study thus far is from Norcross et al. [47], who observed slightly less infiltrating PMNs and macrophages in the rabbit conjunctiva infected with a non-encapsulated *S. pneumoniae* strain than eyes infected with an encapsulated strain. Specifically, rabbit conjunctivae infected with the encapsulated strain had more infiltrating PMNs and macrophages in the bulbar and palpebral conjunctivae compared to those infected with the non-encapsulated strain. Numbers of macrophages and granulocytes peaked at 24 h postinfection in the palpebral conjunctivae and at 48 h postinfection in bulbar conjunctivae for both strains. The authors speculated that this may be due to the infiltration of circulating PMNs first into the palpebral conjunctivae and then into the bulbar conjunctivae. Overall, the capsule did not seem to contribute to the conjunctivitis severity [47].

Another *S. pneumoniae* virulence factor is the cholesterol-dependent cytolysin, pneumolysin (PLY). To date, there have been no studies regarding the effects of PLY in *S. pneumoniae* conjunctivitis. Johnson et al. [48] first described PLY as a membrane-damaging, pore-forming toxin that stimulates proinflammatory interactions with human PMNs. In this study, human PMNs exposed to the PLY showed inhibited chemotaxis, increased cell death, and lysis [48]. Later studies reported that PLY was able to cause an influx of Ca^2+^ and increased phospholipase A2 activity and CR3 expression in human PMNs, which were associated with enhanced superoxide production and the release of elastase [49]. Whether PLY is involved in altering PMN activity during pneumococcal conjunctival infection is an open question.

Valentino et al. [50] conducted a comparative genomic investigation of 271 conjunctivitis-causing pneumococcal strains from the United States and reported that most of the conjunctivitis strains were closely related unencapsulated strains. These strains have differing cohorts of pneumococcal virulence factors and the inability to metabolize fucose, which is a sugar present in corneal epithelial cells. Fucose residues are associated in the adhesion of *E. coli* and *P. aeruginosa* to ocular epithelial cells [51]. Incubation with exogenous fucose was reported to suppress inflammation in rabbit corneas and human explanted models of corneal wound healing [52]. The study by Valentino et al. [50] further suggested that these strains may have a growth advantage by conserving ocular fucose and helping sustain an anti-inflammatory environment. Again, whether PMN are affected in this environment is an open question.

### 2.3. Chlamydia trachomatis Conjunctivitis

*C. trachomatis* is an obligate intracellular bacterium that causes trachoma, the world’s leading cause of preventable blindness [53]. This pathogen can infect the conjunctiva, causing three different syndromes: adult and neonatal inclusion conjunctivitis, lymphogranuloma venereum, and trachoma. *C. trachomatis* is the most common agent isolated from cases of chronic follicular conjunctivitis, and is responsible for approximately 20% of acute conjunctivitis cases [54]. Several studies have shown that PMNs are part of the host’s response to eradicate this pathogen, but this response also causes host tissue damage [26,55,56].

Rank et al. [56] used transmission electron microscopy in a model of ocular chlamydial infection in guinea pigs to show PMNs in close proximity with infected mucosal epithelial cells. Importantly, the authors demonstrated that PMNs disturbed epithelial focal adhesions. This suggested that PMNs might cause the release of epithelial cells from the conjunctival mucosal epithelium, effectively damaging the host tissue (Figure 3). Lacy et al. [26] showed that PMNs contribute to host conjunctival tissue damage during chlamydial conjunctivitis in PMN-depleted guinea pigs. PMN depletion decreased pathology, but did not eliminate the damage, suggesting that other cells or factors contributing to this pathology. While PMNs may not be essential for direct chlamydial clearance in this conjunctivitis model, PMNs might help modulate the adaptive response by downregulating humoral immunity and promoting T-cell recruitment [26].

Lacy et al. [26] showed that PMNs may downregulate IgA humoral responses in chlamydial infection of the eye. The authors suggested that PMN contributes to the downregulation of TGF-β and IL-5, since these mediators were increased when PMNs are depleted. TGF-β and IL-5 were both required for the production of IgA. How PMNs downregulate these two cytokines is uncertain. However, TGF-β may play an important role because of its multiple functions during an immune response, including the downregulation of inflammation and promotion of IgA production [57]. The evidence presented in this study suggested a greater intricacy in the PMN response to chlamydial infection in the eye than previously suggested. Significantly, this study was the first to propose that PMNs may shape antichlamydial adaptive responses, but may not be important in directly killing chlamydiae at the same time as host tissue damage is occurring [26].

### 2.4. Conjunctivitis Conclusions

Although most cases of conjunctivitis are benign with a self-limited process, this infection can be severe and threaten sight. The standard therapy for conjunctivitis continues to be antibiotics, regardless of the causative agent. Topical antibiotics, such as ciprofloxacin, have been shown to reduce the time of infection, decrease transmissibility, and accelerate recovery [22,58]. Topical corticosteroids are not suggested for bacterial conjunctivitis, although the inflammation can cause discomfort [59]. The conjunctivitis studies discussed above demonstrate that one of the major contributors of this inflammation is the influx of PMNs. The amount of PMN influx has been shown to be dependent on the virulence of the pathogen, and in the case of *S. aureus* infections, dependent on the expression of a polysaccharide capsule. The presence of a capsule on *S. pneumoniae* does not seem to matter in conjunctivitis infections. It is clear that future studies should focus on factors that contribute to PMN responses in *S. pneumoniae* conjunctivitis. *C. trachomatis* conjunctivitis models have revealed that PMNs are partly responsible for epithelial damage and the downregulation of humoral immune responses, which drives an adaptive immune response to clear the infection. Studies on mechanisms to reduce humoral responses and increase adaptive responses in this disease may be helpful. Overall, these bacterial conjunctivitis studies reveal a trend that will be seen throughout this review: PMN recruitment into the eye is mainly dependent on the expression of bacterial virulence factors, and the PMN response is damaging to the host cells of the eye.

## 3. Keratitis

Keratitis is a potentially sight-threatening ocular disease, which may result from injuries and epithelial defects of the cornea. The exposure of damaged corneal epithelium to pathogenic bacteria can lead to inflammation of the layers of the cornea, or keratitis. A healthy and intact ocular surface prevents most pathogens from causing infection, but once corneal epithelial barriers are breached and infected, host defenses act to clear infection. PMNs comprise a significant portion of this initial host immune response, and have been shown to have a major role in influencing the outcome of infection [60]. This host immune response, along with pathogens invading the corneal stroma, can lead to a loss of vision due to corneal scarring [60,61,62,63]. Predisposing factors of susceptibility to keratitis include the misuse of contact lenses and their sterilizing solutions, ocular surgery or other trauma, chronic ocular surface disease, or systemic diseases such as diabetes mellitus [60,64]. Patients with infectious keratitis commonly present with tearing, redness, pain, and blurred vision. However, the clinical presentation, and subsequently the PMN response, largely varies regarding the bacteria responsible for inducing the infection.

In temperate climates, approximately 90% of keratitis cases are caused by bacteria. However, microbial keratitis accounts for 60% of cases in subtropical climates, and fungal keratitis accounts for 35% of cases in tropical climates [65]. Bacterial keratitis is most often associated with contact lens use in the U.S. These severe infections can cause permanent vision loss, which requires corneal transplantation [66]. The most common organisms that cause bacterial keratitis include *Pseudomonas aeruginosa*, *Staphylococcus aureus*, and *Streptococcus pneumoniae* (Figure 4) [60]. Some bacteria initiate infection by contacting the host cell-surface receptors using adhesins. Adhesins, such as pili or fimbriae, facilitate binding to corneal epithelial cells, and may act as toxins, disrupting barriers, initiating microbial invasion, and activating inflammatory cascades [67,68]. The adherence to the damaged corneal epithelium of *P. aeruginosa*, *S. pneumoniae*, and *S. aureus* is significantly greater compared to that of other bacteria, which may explain their frequency in isolation from keratitis cases [69,70]. Once invasion has ensued, corneal tissue can be quickly damaged due to the activities of bacterial toxins and proteolytic enzymes, the activation of corneal metalloproteases, and stimulation of the immune response [62,71]. Responding PMNs also contribute to corneal damage by releasing reactive oxygen species (ROS) and host proteases. These responses are also important for clearing the infection, but some bacteria are able to avoid PMN killing by modulating the antimicrobial functions of PMN in the cornea [72].

### 3.1. Pseudomonas aeruginosa Keratitis

Early research in the 1970s showed that PMNs are the predominant cell type composing the exudate caused by *P. aeruginosa* cornea infection [63,76]. Chusid et al. [77] examined the role of PMN in innate immune resistance to *P. aeruginosa* keratitis in neutropenic guinea pigs and found that fewer PMNs infiltrating into the cornea lead to a larger bacterial burden in the eye and less corneal edema. The PMN response to this infection may have also been important for preventing lethal sepsis, as shown in a study using *P*. *aeruginosa*-infected and cyclophosphamide-treated mice [78]. Interestingly, PMNs were critical for preventing the spread of *P. aeruginosa* to the brain, possibly via the optic nerve. This observation was also observed in corneal infections in Myeloid Differentiation primary response 88 (MyD88)-deficient mice, in which PMN recruitment is defective. MyD88 is a key signal transduction molecule that mediates the activation of cells after Toll-like receptor (TLR) and/or IL-1 and TNF receptor stimulation [79]. MyD88-deficient mice had functional PMNs in the blood, but these infected mice had observable amounts of *P. aeruginosa* in the brain after the induction of keratitis, suggesting a non-vascular route for bacteria to the brain. A similar observation was made more recently, with the discovery that PMN NETs are important in preventing the spread of *P. aeruginosa* to the brain after corneal infection. Thanabalasuriar et al. [80] reported that dissemination to the brain was prevented by a PMN NET barrier generated in response to the *P. aeruginosa* expression of the type-3 secretion system (T3SS) in a biofilm. Indeed, the T3SS in the Gram-negative *P. aeruginosa* seems to be important for modulating PMN behavior during corneal infection. Effector proteins secreted by T3SS, such as ExoS and ExoT (Exoenzyme S and T), are responsible in promoting PMN apoptosis in *P. aeruginosa* keratitis [81] and inhibiting reactive oxygen species production in neutrophils [72].

The flagellum of *P. aeruginosa* also activates innate immune responses in corneal epithelium through interaction with TLRs, which initiate innate immune cascades leading to the production of proinflammatory cytokines in the cornea. In a study conducted on immortalized human corneal epithelial cells (HCECs), Zhang et al. [82] observed that *P. aeruginosa* flagellum could signal the NF-κB system through TLR5 by inducing phosphorylation and the degradation of IκB-α (a regulatory protein that inhibits NF-κB). This led to the expression and secretion of proinflammatory cytokines IL-6 and IL-8, which are both important in regulating PMN infiltration. In addition to IL-6 and IL-8, the chemokine IL-1β is also a critical mediator of the innate host response to *P. aeruginosa* keratitis. By using IL-1β-deficient mice, Karmakar et al. [83] demonstrated that IL-1β was essential for PMN recruitment and bacterial clearance in *P. aeruginosa* keratitis. PMNs were the primary source of IL-1β in vivo during this infection, and IL-1β cleavage during infection was dependent on neutrophil elastase, which is a serine protease. This cleavage led to a greater infiltration of PMNs, which was beneficial for bacterial clearance and the prevention of dissemination, but detrimental to the clarity of the corneal epithelium.

Other chemokines also play an important role in *P. aeruginosa* keratitis. Xue et al. [84] showed that the CCL2 and CCL3 were critical in recruiting PMNs to the cornea. Treating mice with anti-CCL2 or anti-CCL3 antibodies caused less corneal damage severity and PMN infiltration compared to control antibody-treated eyes. However, antibody treatment did not change the rate of bacterial clearance from the cornea. These results support the contention that CCL2 and CCL3 are important regulators of PMN recruitment, which may lead to therapies that target CCL2 and CCL3 in the treatment of *P. aeruginosa* or possibly other forms of bacterial keratitis.

### 3.2. Staphylococcus aureus Keratitis

A complex PMN response is shared amongst many microbial ocular infections, especially with *S. aureus* keratitis, which is the most common cause of bacterial keratitis worldwide [30,60,64,85]. Unfortunately, an increased incidence of corneal infections by MRSA has also emerged [86,87]. When *S. aureus* infects the corneal stroma and epithelium, it quickly replicates and produces toxins such as hemolytic α-toxin [88]. This results in tissue damage, epithelial ulceration, and possibly corneal opacity. *S. aureus* keratitis also results in PMN infiltration to the corneal stroma. The PMN response of degranulation and the release of cytotoxic mediators possibly contributes to the pathogenesis of this disease [88,89,90,91,92].

Corneal epithelial cells recognize Gram-positive bacteria via the activation of TLRs. This leads to the influx of PMNs to the site of *S. aureus* keratitis, which has been linked to the activation of TLR2 by the bacteria. Human corneal epithelial cells (HCECs) express TLR2 and, in vitro, respond to viable *S. aureus*, its secreted proteins, and peptidoglycan, but not lipoteichoic acid, by triggering the activation of mitogen-activated protein kinase (MAPK) and NF-κB signaling pathways. Importantly, HCECs also expressed and secreted proinflammatory cytokines, such as IL-6, IL-8, and TNF-α. These mediators can recruit inflammatory PMNs to the site of infection [93]. These in vitro observations were supported by a study using TLR2-deficient and MyD88-deficient mice in which TLR2/MyD88 functioned as a detector of *S. aureus* in the cornea and mediated infiltration of PMNs [94]. The chemokine receptor 2 (CXCR2) has also been implicated as a facilitator of the inflammatory response during *S. aureus* keratitis. Cole et al. [95] observed that an absence of CXCR2 in mice led to reduced PMN infiltration and higher bacterial replication in eyes compared to that in wild-type (WT) mice, even when chemokines were more highly produced. CXCR2-deficient mice had higher expression levels of ICAM-1 in corneas compared to those in WT mice. Thus, the authors suggested that CXCR2-mediated signaling via the upregulation of adhesion molecules was vital to vascular PMN margination in this model. Khan et al. [96] also reported similar results in a mouse *P. aeruginosa* keratitis model. The authors concluded that the infiltration of PMNs into the corneal epithelium during *P. aeruginosa* and *S. aureus* keratitis was highly dependent on IL-8 activating CXCR2, which therefore upregulated adhesion molecules that are needed for PMN infiltration.

Surfactant protein D (SP-D) is important in host defense and innate immunity, and its role has been studied in *P. aeruginosa* [97,98] and *S. aureus* keratitis. This surfactant-associated protein is an innate immune molecule that is capable of binding to lipids and carbohydrates on the surfaces of microorganisms. Importantly, SP-D also binds to receptors on the surface of phagocytic and inflammatory cells, and acts as an opsonin to increase the rate of microbial clearance. In a mouse model of *S. aureus* infection, Zhang et al. [99] injected the eyes of WT and SP-D-deficient mice with *S. aureus* and in the presence or absence of a cysteine protease inhibitor (E64), which reduced the degradation of SP-D by the *S. aureus* cysteine proteases. Bacterial phagocytosis by PMNs was increased in WT mice compared to that of SP-D deficient mice (Figure 5), and WT mice had reduced ocular injury compared with that of SP-D deficient mice. When cysteine inhibitor was present, the WT mice had greater bacterial clearance and reduced ocular injury compared to that of SP-D-deficient mice [99]. Thus, this data suggested that although SP-D protected the ocular surface from *S. aureus* infection, *S. aureus* cysteine proteases impaired SP-D function. The authors suggested that for *S. aureus* keratitis, a cysteine protease inhibitor may be a potential therapeutic agent.

Once PMNs arrive to the site of *S. aureus* corneal infection, these cells are susceptible to the activities of toxins produced by the organism. Approximately 95% of ocular *S. aureus* isolates carry the *hla/hly* gene and have been reported to produce α-toxin [100,101]. α-Toxin subunits bind to and enter the cellular cytoplasmic membrane and oligomerize into a circular pore, facilitating cellular dysregulation. α-Toxin binding also leads to the cleavage of E-cadherin molecules, altering barrier function [102,103]. Whether this activity extends to the cornea in keratitis is an open question. α-Toxin’s implications on PMN activity during *S. aureus* keratitis have been studied to some degree. Callegan et al. [88] reported that rabbit corneas infected with *S. aureus* lacking α-toxin did not have as many infiltrating PMNs compared to rabbit corneas infected with WT *S. aureus*. *S. aureus* also produces β-toxin, a sphingomyelinase, which was reported to have a minimal contribution to the inflammation observed in a rabbit keratitis model [91]. Gamma-toxin, another pore-forming toxin produced by *S. aureus*, has been reported to contribute to keratitis inflammation and virulence, but not to the extent as that of α-toxin [104].

The role of *S. aureus* protein A, an immunoglobulin-binding cell wall-associated exoprotein, as a virulence factor in *S. aureus* in corneal infection, has also been analyzed. In vivo, the absence of protein A did not affect inflammation in rabbit corneal infections [88]. An in vitro analysis of HCECs indicated that purified protein A induced an inflammatory response via the secretion of chemokines and proinflammatory cytokines via a mechanism separate from that of the activation of TLRs [105].

### 3.3. Streptococcus pneumoniae Keratitis

*Streptococcus pneumoniae* has been reported as the third-leading cause of bacterial keratitis, after *P. aeruginosa* and/or *S. aureus* [64,106,107]. In one study, keratitis caused by *S. pneumoniae* accounted for 33.3% of all bacterial keratitis cases [108]. Treating this infection has become increasingly more challenging because of increasing *S. pneumoniae* resistance to antibiotics [108,109,110]. Keratitis caused by *S. pneumoniae* is not usually associated with the use of contact lenses, as are *S. aureus* and *P. aeruginosa*. Instead, the predisposing factors of *S. pneumoniae* keratitis include ocular trauma or surgery [111,112,113,114]. The corneal damage observed in pneumococcal keratitis has been credited mainly to pneumococcal virulence factors, such as the toxin pneumolysin (PLY), which damage cells and initiate a robust immune response [11,71,115,116,117,118,119].

The role of PLY in keratitis was first analyzed in rabbit models. Rabbits intrastromally infected with PLY-defective strains of *S. pneumoniae* had reduced corneal virulence compared to rabbits infected with WT strains [115,117,119]. Norcross et al. [117] reported similar results with primary rabbit corneal epithelial (RCE) cells in an in vitro model of pneumococcal keratitis. PLY seems to play a role in disease severity by triggering an inflammatory response during corneal infection. During pneumococcal keratitis, fewer PMNs infiltrated to the site of infection in rabbit eyes infected with a PLY-deficient strain compared to that of WT-infected eyes [117]. Leukopenia in rabbits resulted in a decreased severity of damage to the cornea following challenge with purified PLY [116]. This suggested that PMNs are an important instigator of corneal damage during infection. A more recent study investigated the mechanisms by which PMNs process IL-1β in response to *S. pneumoniae* keratitis. In a mouse model, Karmaker et al. [120] reported that PMNs were the predominant source of IL-1β production, which was dependent on PLY triggering the NLRP3/ASC inflammasome and caspase-1. Therefore, these studies indicated that PLY was at least partly responsible for activating the inflammatory response and causing immune-mediated damage.

Another well-studied virulence factor of *S. pneumoniae* in keratitis is its polysaccharide capsule. The capsule allows bacteria to avoid the host immune system by prohibiting contact between complement components and their receptors on phagocytic cells, preventing killing [121]. Most pneumococcal keratitis cases are reportedly caused by encapsulated bacteria, but specific capsular serotypes are seldom reported [122,123,124]. Reed et al. [125] reported that the capsule was not an important virulence factor in *S. pneumoniae* keratitis in the rabbit, but was essential for virulence in intraperitoneal infections in the mouse. When rabbit corneas were infected with a non-encapsulated *S. pneumoniae*, bacterial growth was less after 48 h postinfection compared to WT infections. The authors speculated that this was due to the PMNs’ ability to more efficiently reduce the numbers of *S. pneumoniae* lacking a capsule [125]. Norcross et al. [47] conducted a similar study in rabbit corneas using a human *S. pneumoniae* keratitis isolate. In this model, the progression of keratitis was unaffected by the absence of the capsule, but the absence of the capsule facilitated faster pneumococcal clearance. Thus, the pneumococcal capsule’s importance in evading phagocytic death in other infections is also important in keratitis.

### 3.4. Keratitis Conclusions

Corneal transparency is important for vision. Bacterial keratitis threatens the clarity of this tissue and ultimately sight when not sufficiently treated at early stages. If antimicrobial treatment is delayed, about 50% of eyes with keratitis gain useful vision [126]. Typical treatments include the use of a combination of topical antibiotics such as cefazolin, tobramycin, and/or gentamicin. Fourth-generation fluoroquinolones, such as gatifloxacin and moxifloxacin, are also good alternatives [127]. However, killing the bacteria with these antibiotics does not completely clear inflammation. The secreted virulence factors and capsule components are still present after bacteria are killed. Therefore, the inflammatory response to these components might still contribute to PMN recruitment. PMNs contribute to host tissue damage in the cornea, which can lead to corneal scarring. However, PMN depletion only results in greater bacterial growth in the cornea, which typically has devastating consequences. Thus, the conundrum of protective and detrimental PMN responses to bacterial infection in the cornea remains.

## 4. Infectious Uveitis Models

Uveitis defines a collection of conditions characterized by intraocular inflammation. Uveitis technically describes the inflammation of the whole eye, but this section will cover models specifically used to study ocular inflammation in which endotoxin is used to mount an immune response. Infectious uveitis is one of the most frequent and devastating causes of blindness worldwide [128,129,130].

Bacterial uveitis caused by *Treponema pallidum*, *Borrelia burgdorferi*, and *Mycobacterium tuberculosis* results in significant intraocular inflammation. The spirochete, *T. pallidum*, causes syphilis. High-risk behavior in HIV and syphilis patients undergoing therapy has added to the increasing occurrence of this disease. Uveitis is the most common ocular manifestation of syphilis, which occurs in approximately 5% of patients with tertiary syphilis [131]. This infection in the eye can occur at any stage of acquired syphilis. *B. burgdorferi* causes Lyme disease, which occurs when this spirochete is transferred during the bite of an *Ixodes* tick. Although rare, ocular manifestations of Lyme disease include uveitis and have been reported during all stages of the disease [132,133]. Tuberculosis uveitis is most commonly caused by *M. tuberculosis*, with this species isolated in 5.6–10.1% of uveitis cases in India where pulmonary tuberculosis is endemic [134,135]. Unfortunately, infectious uveitis can result in visual loss if the disease is unrecognized or treated incorrectly as non-infectious ocular inflammation. The inflammatory process is divided into acute and chronic inflammation. During acute inflammation, the primary infiltrating cells are PMNs and macrophages. Edema and vascular dilation and congestion also occur. In chronic inflammation, the main infiltrating cells in are lymphocytes and macrophages. PMN responses in bacterial uveitis caused by the pathogens listed above have not yet been explored in animal models.

### 4.1. Endotoxin-Induced Uveitis

Endotoxin-induced uveitis (EIU) is an animal model that is used for studying the inflammatory mechanisms in infectious uveitis. EIU is initiated by using non-antigen-specific stimuli. This model also serves as a useful example of human uveitis. The lipid A moiety of bacterial endotoxin causes biological activity and effects such as hypotension and fever. Endotoxin as an inducer of uveitis was first used in 1943 when Ayo [136] demonstrated that a lone intravenous injection of endotoxin could cause inflammation in the eye in large laboratory animals.

Kinetic studies in mice showed that PMNs first migrated into the eye at approximately 6 h after endotoxin injection. Ocular inflammation peaked about 18 h later, suggesting that these PMNs were the main contributing factor to uveitis inflammation. Whitcup et al. [137] reported high Mac-1 expression on infiltrating PMNs and mononuclear cells 12 h after endotoxin injection. In *Salmonella typhimurium* endotoxin-injected mice, treatment with an anti-Mac1 antibody greatly reduced inflammatory cell infiltration in the uvea and lowered the inflammation grade compared to inflammation in control mice that were not treated with the antibody [138]. Li et al. [139] observed that caveolin-1, a protein of caveolae membrane microdomains, is involved in PMN recruitment in inflammation in EIU. This group measured the number of PMNs infiltrating to the intraocular space following the injection of *S. typhimurium* LPS in WT and caveolin-1 (Cav-1)-deficient mice. Cav-1 deficiency caused a significantly increased recruitment of immune cells and increased leukostasis compared with controls (Figure 6). The authors hypothesized that the Cav-1 deficiency rendered the retinal vasculature more permeable, since Cav-1 is a component of transendothelial and trans-RPE pores, which promote immune cell transmigration [139].

Inflammation in EIU is linked to the release of cytokines, such as TNF-α, interleukin-l (IL-l), IL-6, and IL-8 [140,141]. Endotoxins also prime PMNs for the release of leukotriene B4, which is important for further recruitment of PMNs, and has been observed in an EIU rat model [142]. Clearly, these studies using the EIU model support PMNs playing a significant role in inducing inflammation during bacterial uveitis.

Interestingly, pre-exposure to an endotoxin such as LPS causes reduced sensitivity to a second LPS challenge. This phenomenon is referred to as endotoxin tolerance. In the EIU model, repeated LPS injections into the footpad causes LPS tolerance and resistance to developing uveitis [143,144]. LPS is chemotactic for PMNs, and after challenging an animal with LPS a second time, PMN infiltration is inhibited dramatically [145]. Chang et al. [146] reported that in patients with acute anterior uveitis, peripheral blood PMNs and monocytes may exhibit endotoxin tolerance. This suggests that there are systemic factors that might be involved in the development of EIU tolerance. To further understand this mechanism, Mashimo et al. [147] used the LPS-tolerant EIU rat model and found that the reduction of peripheral blood PMN chemotaxis and constant high expression of IL-10 in the eye contributes to LPS tolerance.

### 4.2. Uveitis Conclusions

The EIU model in rodents mimics many immunopathogenic mechanisms that are associated with human uveitis, which is important in extrapolating data to human disease, and testing and developing novel therapies. Many EIU protocols utilize mouse models. This includes LPS injections into the eye, tail vein, footpad, and skin. The LPS-induced EIU model conveniently induces EIU within hours in mice, which makes this model appropriate for the studying basic mechanisms and possible therapeutic strategies. These models have been useful in studying PMN responses, such as migration and ROS release, in uveitis [139,142]. Some limitations to this model are the varied injection methods and routes, as well as the use of LPS preparations of different bacteria that may generate inconsistencies in experimental outcomes, and questions when extrapolating data from one uveitis model to another.

## 5. Endophthalmitis

Bacterial endophthalmitis is an infection and inflammation that occurs when microorganisms are introduced into the posterior segment of the eye. Endophthalmitis may follow intraocular surgery (postoperative), a penetrating injury to the eye (post-traumatic), or from metastatic spread of bacteria into the eye from a different anatomical site (endogenous). During infection, irreversible damage to retinal tissues frequently occurs. Inflammation and vision loss are devastating consequences of this infection (Figure 7).

The incidence of endophthalmitis after trauma has been estimated to occur in 3–17% of cases [33,148,149,150]. After cataract surgeries, which are the most common type of ocular surgery performed, the occurrence rate ranges from 0.056% to 0.57% [151]. Despite the low rate, this infection poses a significant health risk due to the large number of cataract surgeries performed each year. The World Health Organization estimates that the number of cataract surgeries will rise to 32 million per year by 2020 [152]. The range of the microorganisms causing endophthalmitis differs in various parts of the world. Gram-positive bacteria, including *Bacillus, Staphylococcus*, *Streptococcus*, *Enterococcus*, and other Gram-positive species cause over 75% of culture positive cases of endophthalmitis in Western countries [153,154], while Gram-negative species comprise only 6% of endophthalmitis cases. In Far Eastern countries, Gram-positive bacteria account for approximately 53% of postoperative cases, and up to 26% may be caused by Gram-negative bacteria [152,153,154,155,156]. Endophthalmitis can also be a complication of keratitis, since keratitis can result in perforation of the cornea and contamination of the interior of the eye with the infecting organism [157].

Ocular damage is caused by both the bacteria and by the immune response. Indeed, bacterial growth and direct toxicity from bacterial products cause damage to host tissue, but the excessive inflammatory response may also be responsible for decreased visual outcome due to retinal toxicity from noxious PMN products. Many retinal cell types do not regenerate, so it is essential to reduce the damage caused by inflammation due to infiltrating PMNs, which are the predominant inflammatory cell type in the eye during the earliest stages of bacterial endophthalmitis [10,161,162,163,164]. The initial inflammatory response is unavoidable and may be the earliest sign of visual disturbance for the patient. Clearance of bacteria from the eye is vital and depends on an intricate host response characterized by the early recruitment of PMNs into the eye. Bacterial clearance by PMNs occurs via a coordinated effort of multiple activities, such as phagocytosis, and the release of reactive oxygen intermediates and granule enzymes (cathepsin G, myeloperoxidase [MPO], lactoferrin, and elastase). These products of PMN activation may cause enhanced tissue damage [165]. Therefore, PMN activities and their products designed to clear infections are likely in part responsible for retinal toxicity and subsequent vision loss in endophthalmitis.

### 5.1. Staphylococcus aureus Endophthalmitis

*S. aureus* is a chief cause of post-traumatic and postoperative endophthalmitis. The visual outcome of the disease is usually poor, and many cases result in final visual acuities of 20/400 or worse [151,166,167,168]. The development of antibiotic resistance in clinical isolates may result in an increased incidence of treatment failures for *S. aureus* endophthalmitis [169,170].

*S. aureus* secretes many extracellular toxins, which include cell wall-associated proteins/adhesins (clumping factor, protein A, and fibrinogen and fibronectin-binding proteins) and extracellular virulence factors (pore-forming toxins, lipases, and proteases). These secreted and cell wall components induce inflammation and likely direct toxicity on important tissues responsible for vision during *S. aureus* endophthalmitis. The effects of staphylococcal factors on intraocular inflammation were initially examined by injecting metabolically inactive bacteria, purified bacterial cell walls, and culture supernatants into rabbit eyes [161]. The injection of metabolically inactive bacteria or cell walls caused less inflammation and infiltration of PMNs compared to eyes injected with live bacteria or cell-free supernatants. Live bacteria and culture supernatants were also more toxic for the retina than cell walls or metabolically inactive bacteria, which suggests that retinal tissue damage was caused by secreted toxin(s) [161]. Individual staphylococcal cell wall components may also be important in driving inflammation in endophthalmitis. Suzuki et al. [171] reported the significant blunting of intraocular inflammation following infection with a *tarO*-deficient *S. aureus* mutant. This mutant was deficient in wall teichoic acids, suggesting that this cell wall component is important in inflammation during *S. aureus* endophthalmitis.

The roles of individual toxins in *S. aureus* endophthalmitis were first examined by comparing infections with mutant strains deficient in α-, β-, or γ-toxin to infections caused by a WT parental strain in a rabbit model of endophthalmitis [148]. Of these toxins, the absence of α-toxin resulted in the preservation of retinal function. The PMN response to infection with these strains was not investigated. Kumar and Kumar [172] later assessed the role of individual *S. aureus* cell wall components and cell surface or secreted proteins in mouse eyes. These included peptidoglycan, lipotechoic acid, heat-killed *S. aureus*, α-toxin, protein A, and toxic shock syndrome toxin 1. Purified virulence factors injected into the eyes of mice induced inflammation and a concentration-dependent release of cytokines and chemokines, including IL-6, IL-1β, TNF-α, KC, and MIP-2 in mouse eyes. This correlated with increased PMN infiltration, vascular leakage, and reduced retinal function (Figure 8) [172]. Although it is not known whether the specific concentrations of toxins injected into mouse eyes are replicated in experimental or human cases of endophthalmitis, this study showed that the injection of these purified toxins into the eye could cause changes similar to those observed during clinical infections. 

Gamma-toxin and Panton–Valentine leukocidin (PVL) are two-component leukotoxins of *S. aureus*. Components of these toxins are able to interact with each other to form hybrid toxins. When these were injected into the rabbit vitreous, significant retinal toxicity occurred. Rabbit eyes injected with PVL alone had an increased infiltration of PMNs [173]. Although these results showed that leukotoxins have significant intraocular inflammatory activity, further studies are needed to determine the effects of *S. aureus* leukotoxins on PMNs during endophthalmitis.

The important role of secreted toxins in virulence during *S. aureus* endophthalmitis is supported by studies examining the genetic regulators that are responsible for the production of secreted proteins. In *S. aureus*, virulence factor expression is regulated by quorum-sensing systems. These regulators function at the transcriptional level, and are termed Agr (accessory gene regulator) and Sar (staphylococcal accessory regulator) [174], among others. In experimental comparisons of *S. aureus* endophthalmitis with WT or global regulatory mutants, regulatory mutant virulence was reduced or absent [164,175,176]. Importantly, there was also less PMN infiltration in the eyes of mice infected with the regulatory mutants [164,175,176]. This suggests that toxins under regulatory control contributed to some degree to the induction of inflammation and infiltration of PMNs. α-toxin has been shown to induce the lysis of leukocytes, such as PMNs, after local injection, and even induce death after systemic injection in animals [177].

The recruitment of PMNs can be modulated during active *S. aureus* endophthalmitis. Giese et al. [178] demonstrated this by treating with anti-PMN antibodies (dAb) in a rat model. Treatment with dAb resulted in a temporary reduction in PMN infiltration, as well as subsequent reduction in intraocular inflammation in the initial course of *S. aureus* endophthalmitis. However, depleting PMNs also resulted in increased numbers of intraocular bacteria. Engelbert and Gilmore [9] reported that FasL, which was thought to negatively regulate the immune response in the eye, actually promoted the clearance of *S. aureus*. When mice deficient in FasL were infected intravitreally with *S. aureus*, the number of recruited granulocytes was decreased compared to infected eyes with FasL. Eyes lacking a functional FasL also had increased bacterial burden and retinal damage [9]. Rajamani et al. [179] investigated the global metabolomic regulation of innate immunity in *S. aureus*-infected mouse eyes. This led this group [180] to investigate the role of AMP-activated protein kinase (AMPK). AMPK is a multi-substrate protein kinase that contributes to regulating various metabolic processes. AMPK is downregulated in *S. aureus*-infected mouse eyes, but restoring its expression reduced the bacterial burden and inflammation in *S. aureus*-infected eyes by preventing NF-kB and MAP kinase signaling. Restoring its expression in vitro also increased PMN phagocytosis and the killing of staphylococci [180]. Retinal transcriptome analysis revealed major inflammatory/immune pathways impacted in a mouse model of *S. aureus* endophthalmitis. JAK/Stat and IL-17A signaling were the most significantly affected [179]. The contributions of those pathways to bacterial endophthalmitis have not yet been addressed, but IL-17 has been shown to induce protective innate immunity against *S. aureus* skin infection and to contribute to the production of antimicrobial peptide/PMN-recruiting chemokines [181,182]. IL-17 has also been studied in fungal keratitis in which PMNs were the cellular source of IL-17 [183,184]. These studies demonstrated the importance of inflammation and inflammatory pathways in *S. aureus* endophthalmitis.

### 5.2. Streptococcus pneumoniae Endophthalmitis

Although *Staphylococcus* species cause the majority of cases of bacterial endophthalmitis, *Streptococcus* species are also a significant cause of infections that result in rapid vision loss [148,151,185]. *S. pneumoniae* has been reported as a main cause of endophthalmitis after ocular surgery, and is one of the main organisms cultured in bleb-associated endophthalmitis [186,187,188,189,190]. Streptococcal species are most often isolated from endophthalmitis cases in patients receiving intravitreal injections when physicians did not utilize facial masks [191]. Unfortunately, the majority of eyes infected with *S. pneumoniae* experience complete vision loss, despite aggressive therapy.

PMNs infiltrate into the eye at approximately 12 h after the injection of *S. pneumoniae* into rabbit eyes [192]. After 48 h postinfection, PMN numbers increased substantially, contributing to ocular damage. As with *S. aureus* endophthalmitis, the virulence factors of *S. pneumoniae* also contribute to endophthalmitis pathology. The polysaccharide capsule and cell wall components have been suggested as important virulence factors in *S. pneumoniae* intraocular infection [193]. In contrast to its limited importance in keratitis, the polysaccharide capsule of *S. pneumoniae* has been shown to be essential for full virulence in endophthalmitis [125,193]. As previously stated, the capsule has a known function of allowing pneumococci to evade phagocytosis [45,193]. In a rabbit endophthalmitis model, a capsule-deficient mutant of a *S. pneumoniae* clinical isolate was compared to its isogenic strain [193]. Although both animal groups had severe infections, less infiltration of inflammatory cells was observed in capsule-deficient mutant-infected eyes compared with eyes infected with the WT strain. The authors noted that the inflammation in the eyes infected with the parent strain was more distinct and damaging than in the mutant-infected eyes. The observation of fewer PMNs in the mutant-infected eyes was reflected by reduced myeloperoxidase (MPO) activity. Interestingly, this study showed significantly more bacteria in WT-infected eyes, which suggested better bacterial clearance in mutant-infected eyes. Together, these data support the argument that the capsule is important for both increasing PMN recruitment and in escaping PMN bacterial clearance during endophthalmitis [194,195]. This group [195] reported that rabbits passively immunized with Pneumovax^®^23 (a pneumococcal capsule-based vaccine) had less PMN in the vitreous compared to rabbits immunized with mock serum. The lack of PMNs in the vitreous resulted in higher bacterial loads in the immunized rabbits than mock-treated rabbits.

Attention has focused on *S. pneumoniae* virulence factors, specifically PLY [192,195,196,197,198]. PLY was first implicated as a virulence factor in a rat model of endophthalmitis in which purified PLY was intravitreally injected into eyes at different doses. Ng et al. [196] reported that purified PLY induced a dose-dependent influx of PMNs and retinal damage. This group [197] compared the virulence of a mutant *S. pneumoniae* strain deficient in PLY activity to a WT strain in a rat endophthalmitis model. Endophthalmitis from *S. pneumoniae* deficient in PLY activity resulted in less inflammation 24 h postinfection compared to an infection from a strain with full PLY activity. However, by 48 h, there was no difference clinically and histologically between PLY-deficient and WT strains, regardless of toxin production. Sanders et al. [199] conducted a similar study in a rabbit endophthalmitis model with clinical *S. pneumoniae* strains that had low or high PLY activity. These results were similar to that in the rat model, in which there was a reduced infiltration of PMNs in eyes infected with a low activity of PLY compared to eyes infected with a high activity of PLY [192]. Sanders et al. [199] also reported that immunizing rabbits with PLY reduced the number of PMN in the vitreous, which resulted in greater bacterial load in the vitreous. Overall, these studies support the idea that PLY contributes to pathogenesis during the early stages of *S. pneumoniae* endophthalmitis.

Autolysin has been studied in *S. pneumoniae* endophthalmitis as well. This enzyme is thought to contribute to meningitis virulence by facilitating the release of inflammatory cell wall components and PLY when cells autolyze [200,201]. An autolysin-deficient strain of *S. pneumoniae* resulted in reduced inflammation and PMN infiltration at 24 h postinfection compared to infection caused by a WT strain in an endophthalmitis model [197]. The direct mechanisms by which PLY and autolysin contribute to inflammation and PMN infiltration in endophthalmitis require further investigation.

### 5.3. Bacillus Endophthalmitis

*Bacillus* is one of the major bacterial pathogens causing post-traumatic endophthalmitis, and is also known for causing endogenous endophthalmitis in intravenous drug abusers [202,203,204]. A majority of *Bacillus* endophthalmitis cases have a rapid course, which usually results in blindness within a few days [203,204]. *Bacillus cereus* is the most common *Bacillus* species isolated from blinding cases of endophthalmitis. *Bacillus* endophthalmitis can also be caused by *Bacillus thuringiensis*, which is a fellow member of the *Bacillus cereus sensu lato* (BCSL) group that is both genetically and phenotypically analogous to *B. cereus* [205]. The hallmarks of *Bacillus* endophthalmitis include rapidly evolving intraocular inflammation, eye pain, a rapid loss of visual acuity, and fever. Fortunately, *Bacillus* species have remained sensitive to currently used ophthalmic antibiotics. However, the rapidly destructive nature of *Bacillus* endophthalmitis calls for immediate and proper treatment, which may include intravitreal injections, systemic antibiotics, and vitrectomy.

As previously discussed, PMNs are a predominant infiltrating cell type that are the first line of defense in innate immunity against intraocular pathogens. In *Bacillus* endophthalmitis in rabbits, PMN were observed in the vitreous in close proximity to the optic nerve at 6 h postinfection, with PMN migrating into the vitreous from the ciliary body shortly thereafter [148]. Similar observations were reported in a mouse model of *Bacillus* endophthalmitis, initiating as early as 4 h postinfection in the same anatomical areas of the eye [10]. Histology, MPO, and flow cytometry confirmed that the main infiltrating cell in experimental *Bacillus* endophthalmitis is the PMN [10].

PMNs function not only as phagocytes, but also synthesize and release chemokines and cytokines, including TNFα [206]. During *Bacillus* intraocular infection, TNFα is detected prior to and during PMN presence in the eye [10], and is important to intraocular pathogen control during experimental *Bacillus* endophthalmitis. Ramadan et al. [207] reported that the absence of TNFα in a mouse model of *Bacillus* endophthalmitis resulted in fewer PMNs migrating into the eye, which resulted in faster retinal function loss and bacterial replication. In this study, IL-6, IL-1β, and MIP-1α were detected during the later stages of infection when large numbers of PMNs were present [206]. To test whether the chemokine CXCL1 or the cytokine IL-6 contributed to PMN recruitment, Parkunan et al. [208] compared *Bacillus* endophthalmitis pathogenesis in WT, IL-6-deficient, and CXCL1-deficient mice. While the absence of IL-6 did not change the overall pathogenesis of endophthalmitis, the absence of CXCL1 resulted in less PMN infiltration and retinal damage. Interestingly, the bacterial burden did not increase in the absence of CXCL1 [208]. This finding is contrary to what has been seen in other ocular infections, such as keratitis, where a reduction of PMNs led to a greater bacterial burden [77].

Similar findings were observed in mice lacking functional innate immune receptor pathways. TLRs, specifically, TLR2 is a Gram-positive pathogen recognition receptor. TLR2-deficient mice with *Bacillus* endophthalmitis had a delayed recruitment of PMN and less inflammation in the eye, which was likely due to the altered expression of recruiting cytokines and chemokines. Similar to the CXCL1-deficient mice, the absence of TLR2 did not change the growth of *Bacillus* in the eye [209]. Although these studies showed reduced inflammation in TLR2-deficient mice, there was residual inflammation, suggesting the contribution of further innate immune recognition and signaling mechanisms in inflammation. These findings spurred on studies by Parkunan et al. [210] to examine the role of TLR4 and its adaptor molecules, TRIF and MyD88, in *Bacillus* endophthalmitis. Although *Bacillus* does not synthesize the canonical TLR4 ligand, LPS, TLR4-deficient, MyD88-deficient, and TRIF-deficient mice each had reduced inflammation and reduced recruitment of PMNs after *Bacillus* intraocular infection. TLR4-deficient mice were also used in a study showing that TLR4 was important for driving the expression of proinflammatory mediators that stimulated acute inflammation and PMN recruitment in *Bacillus* endophthalmitis [211]. These findings suggested a possible benefit in targeting CXCL1, TLR2, and/or TLR4 to control inflammation during *Bacillus* endophthalmitis and possibly other bacterial intraocular infections.

*Bacillus* has several virulence factors, many of which are expressed during endophthalmitis, such as cell wall components, hemolysins, phospholipases, and proteases. Individual toxins have been analyzed using mutants deficient in those toxins in experimental models of *Bacillus* endophthalmitis [162,163,212]. *Bacillus* toxins contribute not only to intraocular damage, but also to acute inflammation via the recruitment of inflammatory cells to the vitreous [10]. Beecher et al. [212] used a rabbit model with purified hemolysin BL (HBL) to show that the injection of this toxin into rabbit eyes caused less PMN infiltration than eyes injected with crude *Bacillus* exotoxin preparations containing many secreted proteins. Later, Callegan et al. [213] used a rabbit model of *Bacillus* endophthalmitis to show that HBL contributed minimally to PMN recruitment. Rabbit eyes infected with an HBL-deficient mutant had similar PMN infiltration compared to that of WT *Bacillus*-infected eyes [213]. Phosphatidylcholine-specific phospholipase C (PC-PLC) was toxic when injected into rabbit eyes [214]. However, Callegan et al. [162] reported that PC-PLC and phosphatidylinositol-specific phospholipase C (PI-PLC) had minimal roles in the recruitment of inflammatory cells during *Bacillus* endophthalmitis. Rabbit eyes infected with PI-PLC-deficient or PC-PLC-deficient mutants had significantly less PMN infiltration at 12 h postinfection, but at 18 h postinfection, the number of PMNs in eyes infected with these mutants was similar to eyes infected with WT *Bacillus* [162]. The expression of most *Bacillus* toxins and enzymes is controlled by a quorum-sensing transcriptional regulator, PlcR. In an endophthalmitis rabbit model, PlcR mutants of *B. cereus* and *B. thuringiensis* were significantly less virulent than WT. The plcR-deficient mutants also delayed the onset of PMN infiltration during infection [215,216]. The reduced virulence in this study was likely due to the reduced expression of virulence factors by plcR-deficient strains. Together, these studies demonstrated the significance of quorum sensing, but perhaps not these individual toxins, to the pathogenicity of *Bacillus* endophthalmitis. Quorum sensing might be thought of as a potential therapeutic target for this disease.

Recently, Mursalin et al. [217] demonstrated that the S-layer protein of *Bacillus* contributed to the intraocular infiltration of PMN during endophthalmitis. Compared to infection with WT *Bacillus*, infection with a strain lacking the S-layer protein SlpA resulted in significantly less MPO and less PMN infiltration in infected mouse eyes. In fact, the SlpA mutation in *Bacillus* resulted in minimal inflammation similar to that observed with WT *Bacillus* infections in TLR2-deficient and TLR4-deficient mice [209,210]. The authors suggested that the S-layer contributed to PMN recruitment by triggering innate inflammatory pathways in the retina [217]. This was the first report of the absence of a single *Bacillus* virulence factor having such a profound impact on the severity of inflammation, suggesting the importance of the S-layer protein as a potential therapeutic target.

### 5.4. Enterococcus faecalis Endophthalmitis

*E. faecalis* is a Gram-positive organism and a human intestinal commensal that is among the leading causes of nosocomial infections [218]. *E. faecalis* is a hazardous bacterium that has acquired resistance to several available antibiotics. As such, *E. faecalis* is ranked seventh among the Centers for Disease Control and Prevention’s (CDC’s) top antibiotic-resistant threats [219]. *E. faecalis* is one of the leading causes of postoperative endophthalmitis, mainly resulting from infected filtering blebs after glaucoma surgery [189,190]. These clinical studies reported that *E. faecalis* is usually associated with a significant loss of vision, with only about 15% of endophthalmitis cases resulting in 20/200 or better visual acuity. The preceding studies indicated that enterococcal virulence factors such as gelatinase, cytolysin, and serine protease contributed to the pathogenesis of *E. faecalis* endophthalmitis [220,221,222,223,224], affecting the recruitment of PMNs to the eye. Stevens et al. [219] conducted the first study documenting an *E. faecalis* endophthalmitis animal model. The authors showed that a plasmid (pAD1) encoding a broad-spectrum cytolysin in *E. faecalis* contributed to the pathogenesis of endophthalmitis. After infecting rabbits with isogenic strains of *E. faecalis* that harbored or lacked a plasmid encoding the cytolysin, eyes infected with the strain lacking the pAD1 had less vitritis and PMN recruitment compared to strains with the plasmid encoding the cytolysin [220]. To further prove that this effect was due to the cytolysin and not any other virulence factors that may have been encoded on the pAD1, Jett et al. [221] tested mutants of *E. faecalis* strains containing Tn*917* transposon insertions in different cytolysin genes in the rabbit endophthalmitis model. Rabbit eyes infected with mutants with Tn*917* inserted into the cytolysin *cylL* gene had markedly less PMN recruitment [221]. Recent studies demonstrated that the activity of cytolysin and resulting intraocular inflammation were reduced following treatment with a biomimetic nanosponge that is capable of binding the large subunit of cytolysin, CylLL [222,225]. In sterile in vivo and in viable *E. faecalis* endophthalmitis mouse models, nanosponge treatment resulted in less inflammation and damage to the eye and preserved retinal function [225,226]. Therefore, the cytolysin is important in inflammation and PMN recruitment, although the mechanism by which cytolysin activates inflammatory pathways remains to be determined.

In *E. faecalis*, the Fsr quorum-sensing system regulates the expression of serine and gelatinase proteases in a cell density-dependent manner. Mylonakis et al. [222] reported that rabbit eyes infected with an *fsrB*-deficient mutant had mild PMN infiltrate into the vitreous, preserved retinal layer structure, and no subretinal inflammatory infiltrate compared to that of WT-infected eyes after 48 h postinfection. Engelbert et al. [222] later showed that rabbit eyes infected with *E. faecalis* serine protease-deficient and gelatinase-deficient mutants had less PMN infiltration and retinal damage than WT-infected rabbit eyes. Suzuki et al. [224] confirmed a role for the *E. faecalis* serine protease in a model of *E. faecalis* endophthalmitis in aphakic rabbits. The authors reported that aphakic rabbit eyes infected with the serine protease-deficient mutant had a delayed PMN influx and significantly less retinal damage. Similar results were observed after injecting culture supernatants from WT or protease-deficient *E. faecalis*. From these studies, it is clear that the Fsr quorum sensing system and the virulence factors it regulates contribute to inflammation and the recruitment of PMNs during *E. faecalis* endophthalmitis.

### 5.5. Gram-negative Bacterial Endophthalmitis

Gram-negative bacteria are more often associated with endogenous endophthalmitis than with other types of endophthalmitis. Gram-negative endophthalmitis has been reported to be more common than Gram-positive endophthalmitis in the Far East, while in Europe and North America, Gram-positive strains were more prevalent in this disease [227,228]. Common Gram-negative organisms isolated from endogenous endophthalmitis cases include *Klebsiella pneumoniae*, *Pseudomonas aeruginosa*, *Escherichia coli*, and *Neisseria meningitidis*. Unfortunately, the visual outcomes connected with these bacterial infections remain poor [227,228].

The most frequently isolated serotypes in endogenous *K. pneumoniae* endophthalmitis are K1 and K2 [229]. K1 serotype strains contain mucoviscosity-associated gene A (*magA*) and regulator of mucoid phenotype (*rmpA*) genes that have been reported to be significant virulence factors in liver abscesses [229,230,231]. From these abscesses, the bacteria can enter the bloodstream and infect the eye, causing endogenous endophthalmitis. Primary underlying *K. pneumoniae* liver abscesses have a 3% to 10% risk for metastatic spread to the eye [232,233]. Two-thirds of *K. pneumoniae* endogenous endophthalmitis patients had a liver abscess caused by the same bacteria, and half were diabetic. Additionally, clinical isolates often express the hypermucoviscous (HMV) phenotype [229,230,231,232,233,234,235]. Even after treatment, a majority of patients with *K. pneumoniae* endophthalmitis lose useful vision [227,228,236,237,238].

In experimental *K. pneumoniae* endophthalmitis, mouse and rabbit eyes intravitreally injected with *K. pneumoniae* had an influx of PMNs from the optic nerve and ciliary body starting at 9 h postinfection [239]. An HMV-deficient *K. pneumoniae* isolate induced less retinal inflammation and function loss compared to eyes infected with a hypermucoviscous isolate. MPO analysis of these eyes suggested greater numbers of PMNs in eyes infected with the hypermucoviscous strain [240]. The significance of MagA in endophthalmitis and PMN recruitment was confirmed by Hunt et al. [241], who reported that mouse eyes infected with WT *K. pneumoniae* had greater PMN influx, which resulted in significantly more retinal function loss than eyes infected with an isogenic *magA*-deficient strain. These authors [242] also showed that TLR4 contributed to PMN recruitment during *K. pneumoniae* endophthalmitis. TLR4-deficient mouse eyes had reduced numbers of PMNs after 12 and 14 h postinfection compared to WT eyes infected with *K. pneumoniae* [242].

*Escherichia coli* and *P. aeruginosa* are also Gram-negative bacteria commonly isolated from endogenous endophthalmitis cases and are also associated with poor visual outcomes [227,228,243,244,245]. Other than the *K. pneumoniae* studies mentioned above, there have been few studies on the contributions of virulence factors to endophthalmitis caused by Gram-negative bacteria. Aizuss et al. [8] reported that an absence of complement via the injection of cobra venom factor in guinea pigs delayed PMN recruitment following the intravitreal injection of *P. aeruginosa*. Astley et al. [239] reported that while inflammation in *E. coli* and *K. pneumoniae* endophthalmitis was severe at 15 h postinfection in mice, inflammation was not as pronounced in rabbit at the same time point postinfection. Overall, the number of studies examining PMN responses in in vivo models of Gram-negative endophthalmitis has been limited.

### 5.6. Endophthalmitis Conclusions

Therapies for bacterial endophthalmitis, including antibiotics and anti-inflammatory drugs, are only effective when given promptly after disease onset to kill microorganisms and limit inflammation and ocular injury [33]. However, current therapies often fail to completely arrest inflammation and do not prevent toxins from damaging tissue and threatening retinal function [71,225,226,239]. Vision loss during endophthalmitis is a clinically relevant problem since there are currently no medical options to repair permanent retinal damage. More effective therapies are needed to mitigate inflammation and toxin production to decrease disease severity and prevent blindness during bacterial endophthalmitis.

From the studies presented above, it may be useful to target the cell wall components of bacteria, since these seem to be important for inducing inflammation by activating TLRs. However, it should be noted that the cell wall components of some bacteria that are important for inflammation in endophthalmitis are not important for inflammation in other ocular infections and vice versa (i.e., protein A of *S. aureus* or the capsule of *S. pneumoniae*). Of course, the models and experimental parameters are different among many studies, which may confound comparisons. Bacterial quorum-sensing mechanisms also seem to be important in the degree of resulting inflammation during endophthalmitis. Targeting these transcriptional regulators instead of single virulence factors may help to reduce the inflammation in these diseases. Targeting important inflammatory pathways on the host side may be beneficial as well. For instance, inducing the expression of AMPK, FasL, or antimicrobial peptides may be helpful in the clearance of bacteria by promoting PMN recruitment and phagocytic activity/bacterial killing, respectively.

## 6. Concluding Remarks

In recent years, a considerable amount of research on the recruitment, regulation, and responses of PMNs has been conducted in different but complementary models of bacterial ocular infections. This work has led to a deeper understanding of the varied biological roles of this inflammatory cell and the bacterial and host factors that influence its recruitment and function (Figure 9) (Table 1). Studies so far strongly suggest that PMN activity is important for bacterial clearance, but in many cases, the mechanisms underlying PMN involvement in ocular infection are not fully understood. The complexity of the immune regulation in the ocular environment does affect PMN activity, and it may be important to target these regulation systems in order to induce a more effective bacterial clearance. These studies also show that targeting bacterial virulence factors is important to mitigate overall inflammation. Many of these virulence factors have properties that are important in circumventing PMN function. Most of the factors discussed in this review also have an importance in inducing PMN recruitment, which could be understood as beneficial since these inflammatory cells are important for bacterial clearance. However, these factors and PMNs also contribute to host tissue damage. To avoid this Pyrrhic Victory scenario, an effective therapy would need to include components that kill bacteria, clear virulence factors that may be left behind, and perhaps induce PMN activity that would clear remaining bacteria with little to no damage to host tissue. Overall, further investigations on the function of PMNs in response to bacterial ocular infection should open new perspectives for a better understanding of the interplay of PMNs with bacteria within the ocular environment.

## Figures and Tables

**Figure 1 microorganisms-07-00537-f001:**
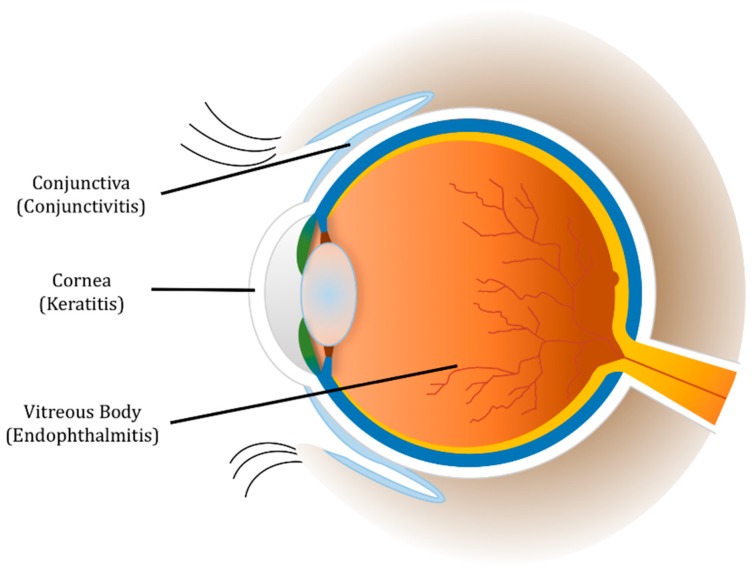
Diagram of the eye and sites of ocular inflammation. This figure illustrates areas of ocular infection and inflammation, such as conjunctivitis, keratitis, and endophthalmitis. The polymorphonuclear leukocyte (PMN) responses to the diseases shown in the diagram will be discussed. Note: Inflammation of the eye is also called uveitis, which will be discussed in its own section.

**Figure 2 microorganisms-07-00537-f002:**
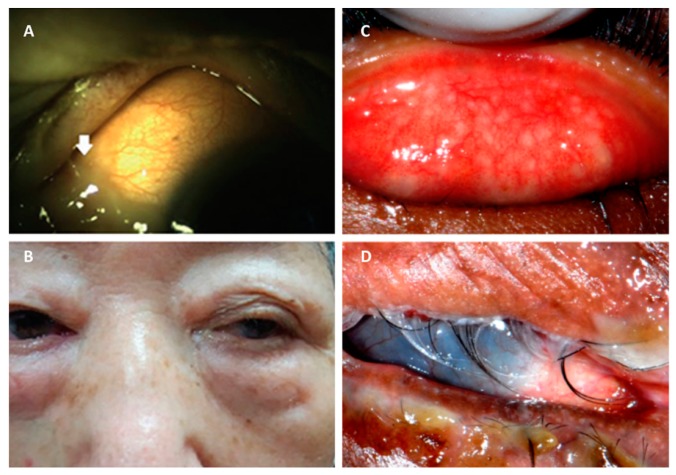
Trachoma and methicillin-resistant *Staphylococcus aureus* (MRSA) conjunctivitis. (**A**) and (**B**) A patient with MRSA conjunctivitis shows thick mucoid discharge at the conjunctival sac (arrow) of the left eye. (**C**) and (**D**) A patient with trachoma with trachomatous inflammation and follicular and/or trachomatous trichiasis (inversion of eyelashes). This figure is a combination of two edited figures reproduced under a Creative Commons License from ©2016 *Korean Journal of Ophthalmology* [27] and © 2013 *PLoS Neglected Tropical Diseases* [28].

**Figure 3 microorganisms-07-00537-f003:**
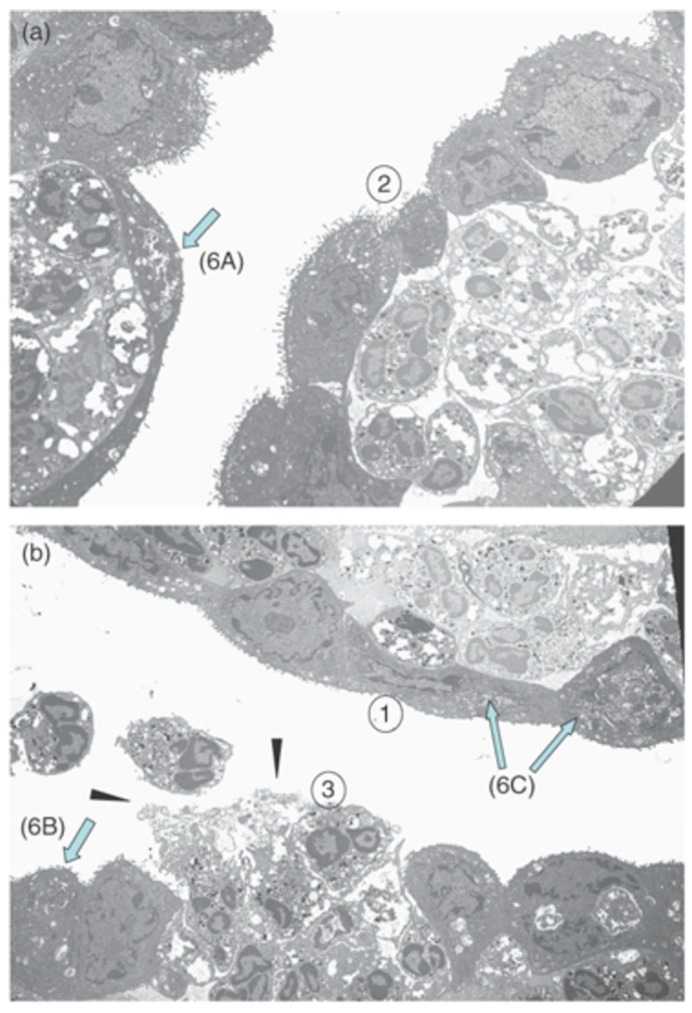
Stages of the PMN response to conjunctival epithelial cells infected by chlamydiae. (**A–B**) (1) PMNs accumulating immediately behind a barrier of infected epithelial cells (arrows). (2) The epithelium loses integrity. (3) PMNs break through the barrier and are released onto the surface, resulting in the release of damaged epithelial cells (arrowheads). The figure is reproduced from Rank et al. [56] with the permission of Oxford University Press.

**Figure 4 microorganisms-07-00537-f004:**
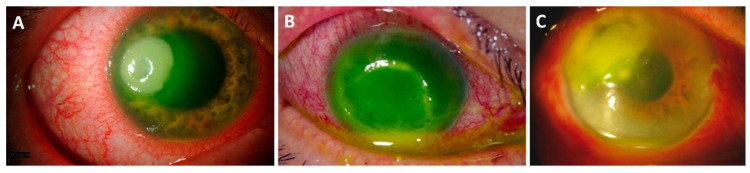
*P. aeruginosa*, MRSA, and *S. pneumoniae* keratitis. (**A**) A corneal ulcer caused by *P. aeruginosa*. PMN fill the ulcer, which may perforate the cornea. (**B**) A patient with MRSA keratitis after penetrating keratoplasty. This patient was treated with topical antibiotics and corticosteroids. (**C**) *S. pneumoniae* keratitis in a patient showing corneal abscess and thinning. Part A is an edited figure reproduced under a Creative Commons Attribution-NonCommercial-NoDerivs 3.0 Unported License from © 2012 *EyeRounds Online Atlas of Ophthalmology* [73]. Parts B and C are reproduced under a Creative Commons Attribution 3.0 License from © 2010 *Korean Journal of Ophthalmology* [74] and © 2009 *Archives of Medicine* [75].

**Figure 5 microorganisms-07-00537-f005:**
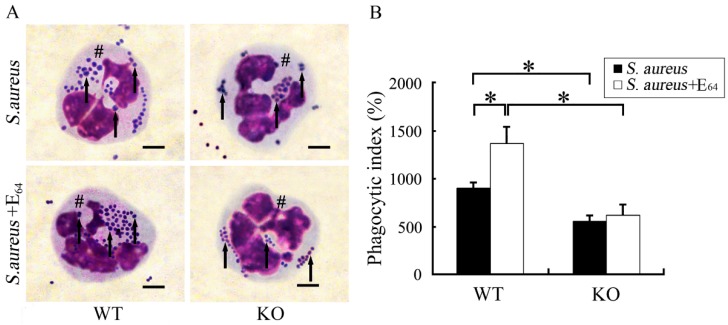
PMN phagocytosis of *S. aureus* is promoted by surfactant protein D (SP-D), and *S. aureus* cysteine protease diminished SP-D activity. (**A**) PMNs shown are from tear fluid after inoculation with *S. aureus* or *S. aureus* with E64 (cysteine protease inhibitor). (**B**) Phagocytic index (PI) from PMNs in the tear fluid of infected wild-type (WT) and SP-D KO mice. #, PMNs; Arrows, *S. aureus*. This figure is reproduced under a Creative Commons License from © 2015 *PLoS ONE* [99].

**Figure 6 microorganisms-07-00537-f006:**
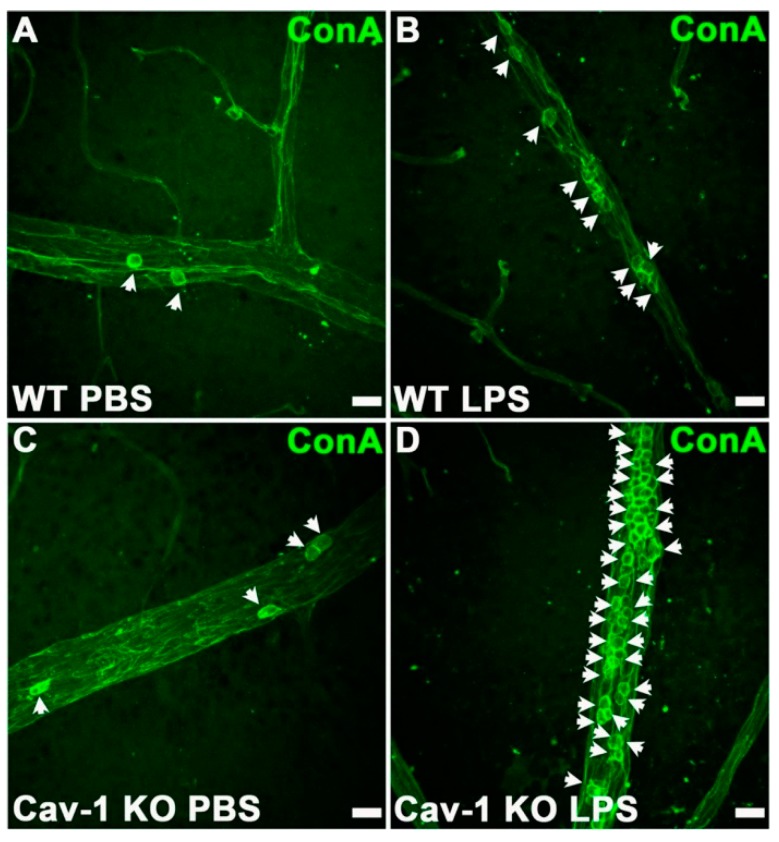
Caveolin-1 deficient mouse retinas show increased leukostasis when challenged with LPS. White arrows point to leukocytes. This figure is reproduced from Li et al. [139] with permission from the Association for Research in Vision and Ophthalmology.

**Figure 7 microorganisms-07-00537-f007:**
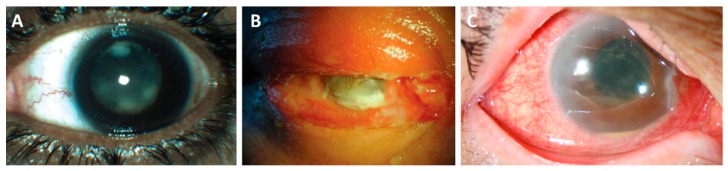
*S.aureus, Bacillus*, and *E. faecalis* endophthalmitis. (**A**) *S. aureus* endogenous endophthalmitis presenting with exudate behind the lens. (**B**) A patient with post-traumatic *Bacillus* endophthalmitis presents with chemosis, corneal opacification, periorbital swelling, proptosis, and a corneal ring abscess. (**C**) A patient with post-operative *E. faecalis* endophthalmitis presents with exudative membrane and infiltrates, ocular injection, and a hypopyon. This figure is reproduced under a Creative Commons Attribution-NonCommercial-ShareAlike License from © 2019 *Indian Journal of Ophthalmology* [158], the Creative Commons Attribution-ShareAlike License 4.0 from © 2017 *Medicine* [159], and the Creative Commons Attribution-NonCommercial 3.0 Unported from © 2015 *J Korean Ophthalmol Soc* [160].

**Figure 8 microorganisms-07-00537-f008:**
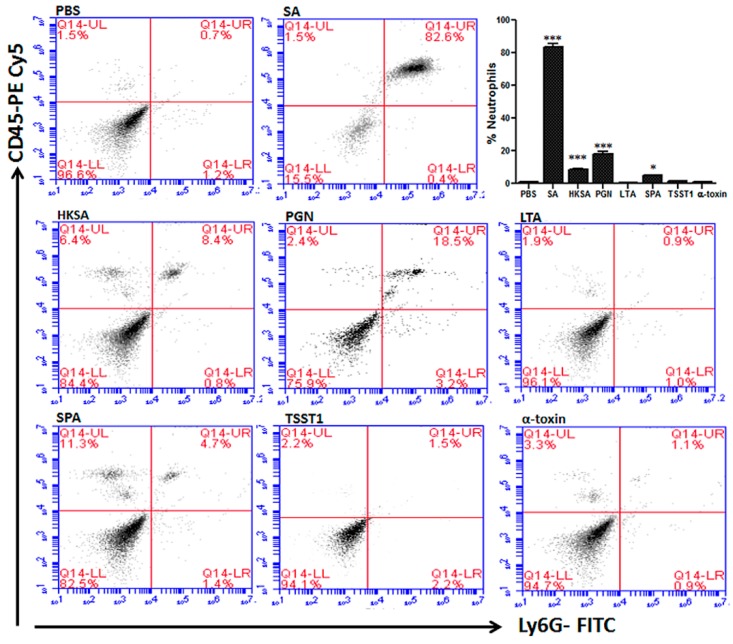
PMN infiltration into mouse retinas following the injection of *S. aureus* virulence factors. Mouse eyes injected with PBS, *S. aureus* (SA), heat-killed *S. aureus* (HKSA), peptidoglycan (PGN), lipotechoic acid (LTA), staphylococcal protein A (SPA), toxic shock syndrome toxin 1 (TSST1), or α-toxin, and flow cytometry was performed to quantify retinal PMNs. This figure from Kumar and Kumar [172] is reproduced under a Creative Commons License from © 2015 *PLoS ONE*.

**Figure 9 microorganisms-07-00537-f009:**
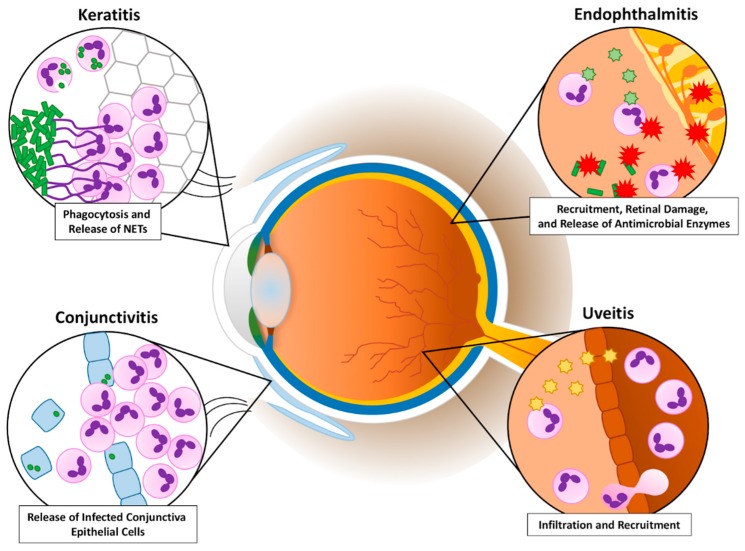
Summary of PMN responses to ocular bacterial infections. This figure illustrates PMN responses that have been observed in the different areas of the eye during conjunctivitis, keratitis, uveitis, and endophthalmitis. During conjunctivitis, PMNs damage the conjunctival epithelial barrier by accumulating and releasing infected epithelia onto the surface of the conjunctiva. The presence of PMNs during this infection causes a decrease in transforming growth factor-beta (TGF-β) and interleukin-5 (IL-5), which is suggested to downregulate IgA humoral responses [26,56]. The PMN response during keratitis includes a release of neutrophil extracellular traps (NETs) to prevent the further dissemination of bacteria. PMNs have also been observed to bacteria in the cornea and produce IL-1β and IL-17 as part of their response [80,83,99,120,183,184]. In some uveitis models, PMNs are recruited quickly into the eye, and self-recruit by producing LTB4 and other proinflammatory molecules [133,142]. PMNs in the endophthalmitis also self-recruit by releasing recruiting chemokines such as TNF-α, but may also cause retinal damage by producing antimicrobial enzymes and reactive oxygen species (ROS) [10,206].

**Table 1 microorganisms-07-00537-t001:** Summary of bacterial components, PMN responses, and inflammatory pathways and cytokines involved in bacterial ocular infections. MyD88: myeloid differentiation primary response 88, SP-D: surfactant protein D, TLR: Toll-like receptor.

Bacterial Pathogen	Bacterial Components	PMN Response	Inflammatory Pathways	Cytokines and Chemokines
**Conjunctivitis**
*S. aureus*	PNAG [37]	Infiltration [25,35,36,37]		
*S. pneumoniae*	Polysaccharide capsule [47,50]	Infiltration [47]		
*C. trachomatis*		Infiltration [26,55,56] Damage to epithelia [26]		TGF-β and IL-5 [26]
**Keratitis**
*P. aeruginosa*	T3SS [80] ExoS and ExoT [72,81] Flagellum [82]	Infiltration [63,76,77,78,79,80,82,84] NETosis [80] Apoptosis [81]	MyD88 [79] NF-κB [82] TLR5 [82] IκB-α [82] SP-D [97,98]	IL-6, IL-8, and 1β [82] CCL2 and CCL3 [84]
*S. aureus*	Peptidoglycan [93] Cysteine proteases [99] α-toxin [88,91,100,101] β-toxin [91] γ-toxin [104] Protein A [88,105]	Infiltration [88,91,94,95,96,104] Phagocytosis [99]	TLR2 [93,94] MyD88 [94] MAPKs [93] NF-κB [93] ICAM-1 [95] SP-D [99]	TNFα [93] IL-6, IL-8, TNF-α [93,96] CXCR2 [95,96]
*S. pneumoniae*	Pneumolysin [11,115,116,117,118,119,120]	Infiltration [116,117] Corneal damage [115]	NLRP3/ASC [120] Caspase-1 [120]	IL-1β [120]
**Endophthalmitis**
*S. aureus*	Lipotechoic acids [171,172] α-toxin [148] β-toxin [148] γ-toxin [148,173] Peptidoglycan [172] Protein A [172] TSST-1 [172] PVL [173]	Infiltration [161,172,173,178]	FasL [9] Complement [9]	IL-6 and 1β [172] TNFα [172] KC [172] MIP2 [172]
*S. pneumoniae*	Polysaccharide capsule [193,199] Pneumolysin [192,196,197,199] Autolysin [197]	Infiltration [192,193]		
*Bacillus*	Hemolysin BL [212,213] PC-PLC [162,214] PI-PLC [162] S-layer [217]	Infiltration [10,148,207]	TLR2 [209] TLR4 [210] MyD88 [210] TRIF [210] NOD2, NLRP3 [239]	TNFα [10,207] IL-6 and IL-1β [207,208,210] MIP-1α [207] CXCL1 [208,210]
*E. faecalis*	Gelatinase [222,224] Cytolysin [220,221,225] Serine protease [222,224]	Infiltration [221,222,223,224]		
*K. pneumoniae*	HMV phenotype [230,236,240,241,242]	Infiltration [238,239,241]	TLR4 [242]	CXCL1, TNF, MIP-1 [242]
*P. aeruginosa*		Infiltration [8]		
*E. coli*		Infiltration [239]

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
