# Peer review of "A Pyrrhic Victory: The PMN Response to Ocular Bacterial Infections"

_microorganisms, 2019, doi:10.3390/microorganisms7110537_

Round 1

Reviewer 1 Report

This review was a nicely written and comprehensive piece regarding the role of PMNs in ocular infectious diseases. Comments are primarily editorial in nature:

Lines 21, 85, and 843: check spelling – “pyrric” or “phyrric” and resolve with title Line 47: “utilized” misspelled Line 75: end of paragraph should have a citation Line 119: remove underline from Staphylococcus aureus Line 174: remove “of” Line 179: remove underline from Chlamydia trachomatis Line 190: add “of” between “release epithelial” Line 258: add “to” between “able avoid” Line 266: “P.” should be italicized Line 269: suggest adding a statement regarding what pathway MyD88 is involved in and define MyD88 Line 274: add space between “was prevented” Line 278: reconcile “flagella” and “activates” i.e. either both singular or both plural Line 279: first instance of TLRs – define here Line 292: remove “of” Lines 307 and 682: may consider not defining TLRs again if defined earlier Line 317: add “to” after “led” Line 335: “…when cysteine inhibitor was present…” not “is” Line 356: suggest also mentioning gamma-toxin Line 383: remove “is” Line 415: “this” instead of “his” Line 424: suggest a period or semi-colon (not a comma) after “disease” Figure 5 legend: suggest stating that the arrows point to leukocytes Lines 475-482: suggest adding a statement regarding PMNs in uveitis overall Line 517: remove underline from Staphylococcus aureus Line 648: remove underline from Bacillus Line 703: “model” instead of “models” Line 731: remove underline from Enterococcus faecalis Line 733: “bacterium” instead of “bacteria” Line 734: suggest defining CDC Line 776: “meningitidis” instead of “meningitis” Line 782: “abscesses” instead of “abscess” Lines 790 and 805: “pneumoniae” instead of “pneumonia” Line 818: “bacteria” instead of “bacterium” Line 849: author contribution of Md Huzzatul Mursalin is missing

Reviewer 2 Report

Following are the comments/suggestions for the authors to improve the current manuscript

It is an interesting review, summarizing the overall role of PMNs in ocular bacterial infections. However, it will be helpful if authors can provide an illustration depicting the beneficial or detrimental role of PMNs in each or overall ocular bacterial infection (s). Also, authors can include a table and describe the role of PMNs briefly (disease model, mechanisms along with references) in each ocular infection. Is there any role of neutrophils subsets (PMN1 and PMN2) in the ocular bacterial infection? If yes, include in the report in the current manuscript.

Minor typos/spelling changes

Pyrrhic spelling different in the title and in line 21, 85, and 843. In line 266, (P. aeruginosa), P should be in italics

Reviewer 3 Report

The authors discuss in details all main processes involved in the bacterial ocular infections, however, reading the text is sometimes difficult to follow.

It would be useful for readers if the authors simplified some of the paragraphs. it could be useful for readers if the authors added summary tables reporting the main characteristics of each pathogen involved in the mechanisms of ocular infection. It could also be useful if the authors added summary diagrams of the main processes of the inflammatory response in each pathological frame highlighting common pathological pathways. As done in figure 2, it could be useful for readers to add some new figures describing different clinical aspects for each different ocular infection.

Reviewer 4 Report

The submitted review by Livingston et al, discusses about the disparate PMN responses to various bacterial infections, specifically S. pneumonia, S. aureus, C. trachomatis and P. aeruginosa infections associated with conjunctivitis and keratitis. The review also discusses the role of endotoxin in uveitis, and provides a very elaborate analysis of endophthalmitis caused by S. aureus, S. pneumonia, Bacillus, E. faecalis, and gram-negative bacteria. This is a very comprehensive review clearly summarizing the specific role of PMNs in pathogen clearance and secondary tissue damage.

Minor comment:

Please check the statement -

Line 245 - “Approximately 90% of all keratitis cases are caused by bacteria [65].” This is not correct if world-wide. In many places of the world, fungal pathogens account for half or more of keratitis cases.

The review is very long and sometimes repetitive, can the conjunctivitis and keratitis section be combined and shortened?

It would also help the readers if the authors created a table of various bacteria, and those components known to trigger inflammation. It could be one comprehensive table for conjunctivitis, keratitis, and endopthalmitis.

Other comments:

Please resolve the typographical errors and the missing articles (grammar), for example,

Lines 13, 174, 190, 274, 317, 383…

line 417 –remove uveitis toward the end of the sentence
